# All-Atomic Molecular Dynamic Studies of Human and *Drosophila* CDK8: Insights into Their Kinase Domains, the LXXLL Motifs, and Drug Binding Site

**DOI:** 10.3390/ijms21207511

**Published:** 2020-10-12

**Authors:** Wu Xu, Xiao-Jun Xie, Ali K. Faust, Mengmeng Liu, Xiao Li, Feng Chen, Ashlin A. Naquin, Avery C. Walton, Peter W. Kishbaugh, Jun-Yuan Ji

**Affiliations:** 1Department of Chemistry, University of Louisiana at Lafayette, P.O. Box 44370, Lafayette, LA 70504, USA; ali.faust1@louisiana.edu (A.K.F.); ashlinnaquin@aol.com (A.A.N.); averycarolinewalton@gmail.com (A.C.W.); peterkish96@gmail.com (P.W.K.); 2Department of Molecular and Cellular Medicine, College of Medicine, Texas A&M University Health Science Center, 8447 Riverside Parkway, Bryan, TX 77807, USA; xjxie@stu.edu.cn (X.-J.X.); mengmengliu@tamu.edu (M.L.); icymoon@tamu.edu (X.L.); 3High Performance Computing, 329 Frey Computing Services Center, Louisiana State University, Baton Rouge, LA 70803, USA; fchen14@lsu.edu

**Keywords:** CDK8, structure, molecular dynamics simulation, kinase domain, LXXLL motif, drug binding site

## Abstract

Cyclin-dependent kinase 8 (CDK8) and its regulatory partner Cyclin C (CycC) play conserved roles in modulating RNA polymerase II (Pol II)-dependent gene expression. To understand the structure and function relations of CDK8, we analyzed the structures of human and *Drosophila* CDK8 proteins using molecular dynamics simulations, combined with functional analyses in *Drosophila*. Specifically, we evaluated the structural differences between hCDK8 and dCDK8 to predict the effects of the LXXLL motif mutation (AQKAA), the P154L mutations, and drug binding on local structures of the CDK8 proteins. First, we have observed that both the LXXLL motif and the kinase activity of CDK8 are required for the normal larval-to-pupal transition in *Drosophila*. Second, our molecular dynamic analyses have revealed that hCDK8 has higher hydrogen bond occupation of His149-Asp151 and Asp151-Asn156 than dCDK8. Third, the substructure of Asp282, Phe283, Arg285, Thr287 and Cys291 can distinguish human and *Drosophila* CDK8 structures. In addition, there are two hydrogen bonds in the LXXLL motif: a lower occupation between L312 and L315, and a relatively higher occupation between L312 and L316. Human CDK8 has higher hydrogen bond occupation between L312 and L316 than dCDK8. Moreover, L312, L315 and L316 in the LXXLL motif of CDK8 have the specific pattern of hydrogen bonds and geometries, which could be crucial for the binding to nuclear receptors. Furthermore, the P154L mutation dramatically decreases the hydrogen bond between L312 and L315 in hCDK8, but not in dCDK8. The mutations of P154L and AQKAA modestly alter the local structures around residues 154. Finally, we identified the inhibitor-induced conformational changes of hCDK8, and our results suggest a structural difference in the drug-binding site between hCDK8 and dCDK8. Taken together, these results provide the structural insights into the roles of the LXXLL motif and the kinase activity of CDK8 in vivo.

## 1. Introduction

CDK8 and CycC are two conserved subunits of the transcription cofactor Mediator complex, which is involved in regulating most, if not all, of RNA Pol II-dependent transcription in eukaryotes [1,2,3,4]. Together with Med12 and Med13, these four proteins form a protein complex known as the CDK8 kinase module, or the CKM [5]. The CKM can physically block the interaction between the small Mediator complex and Pol II [6]. Because CDK8 is the only enzymatic subunit of the Mediator complex, CDK8 can also regulate Pol II-dependent gene expression by directly phosphorylating a number of transcription factors, including E2F1 [7,8], N-ICD (intracellular domain of Notch) [9], p53 [10], Smad proteins [11,12], SREBP (sterol regulatory element-binding protein) [13], and STAT1 (signal transducer and activator of transcription 1) [14,15,16]. Given that amplification or mutations of CDK8 have been identified in a variety of human cancers, such as colorectal cancer and lung neuroendocrine carcinoma [16,17,18,19,20], it is important to understand the function and regulation of CDK8 in different biological and pathological contexts.

Because most of the Mediator subunits being conserved in eukaryotes, we use *Drosophila* as a model organism to explore the function and regulation of CDK8-CycC in vivo. By analyzing the phenotypes of the *Cdk8* and *CycC* mutants during *Drosophila* development, we have previously reported that CDK8-CycC serves as a regulatory node linking dietary perturbations to lipid metabolism and developmental timing during the larval-to-pupal transition [21,22]. Specifically, CDK8-CycC regulates the timing for the larval-to-pupal transition by activating ecdysone receptor (EcR)-dependent gene expression [21]. CDK8 interacts with EcR in both in-vitro and in-vivo assays, and our yeast-two hybrid assay suggests that CDK8 directly binds to the activating function domain 1 (AF1), but not AF2, of EcR [21].

Interestingly, our sequence analysis revealed a highly conserved LXXLL (where L is Leu and X is any amino acid) motif in CDK8 proteins in species ranging from yeasts to humans [21]. The Leu-rich LXXLL motif is often found in transcriptional cofactors of nuclear hormone receptors [23]. For example, the LXXLL motif of the Mediator subunits, MED1 and MED14, are important for their interaction with nuclear receptors, such as androgen receptor, estrogen receptor, glucocorticoid receptor, thyroid hormone receptor, and retinoid X receptor in mammals [23]. *Drosophila* Med14 also has a conserved Leu-rich LXXLL motif and can directly interact with the AF1 domain of EcR [21]. However, the structural properties of the Leu-rich LQKLL motif of CDK8 proteins and the potential structural and functional impacts of mutating it into AQKAA remain unknown.

Similar to other protein kinases, the ATP binding pocket of CDK8 plays a critical role in phosphorylating its substrates. The P154L point mutation within the ATP binding pocket of CDK8 in *Drosophila* eye disrupts the neuronal circuit wiring in the *Drosophila* visual system, and this mutation was predicted to disrupt the CDK8 kinase activity [24]. Given that proteins are highly dynamic and can constantly switch between different conformations, illustrating the structural impacts of both AQKAA and P154L mutants of CDK8 protein may help our understanding of the functional impacts of these mutants in vivo.

Molecular dynamics (MD) simulations can provide atomically detailed views of protein motions in a vacuum or in solutions, sampling multiple timescales ranging from nanoseconds to milliseconds using supercomputing resources [25]. Previously, we used the MD simulations to analyze dynamics of hCDK8, and we have focused on the A-loop and two D173A and D189N mutants within the A-loop [26]. To extend these analyses, we used the CDK8 crystal structure (PDB ID: 3RGF) as the template [27], and performed MD simulations on both the wild-type and mutant CDK8 of human and *Drosophila*. Our main objectives are to characterize the structural differences between hCDK8 and dCDK8, and to study the effect of P154L and AQKAA mutations on global and local structures, which may provide structural insights into the mutant phenotypes that we have observed in *Drosophila*.

As summarized in this work, we have analyzed phenotypes of two mutant forms of CDK8, mutating LQKLL to AQKAA, and P154 to L154, in *Drosophila*. In addition, we have identified unique substructures that distinguish between human and *Drosophila* CDK8 structures. Furthermore, our structure comparison method can nearly accurately cluster proteins with CDK8 point mutations. These results will advance our understanding of the structure-function relationships of CDK8 proteins.

## 2. Results

### 2.1. The LXXLL Motif of CDK8 is Required for CDK8 to Regulate the Timing of Pupariation

We have previously reported that EcR and USP can co-immunoprecipitate with CDK8, that CDK8 can directly interact with the EcR-AF1 domain based on a yeast-two-hybrid assay, and that CDK8 has a highly conserved LXXLL motif [21]. This Leu-rich motif is often found in cofactors that directly interact with nuclear receptors [28]. These observations prompted us to propose that CDK8 may function as a transcription cofactor for EcR, likely through the LXXLL motif [21]. However, no experimental evidence for the functional relevance of either the LXXLL motif or the kinase activity of dCDK8 in regulating developmental transitions are available to date.

Because the wild-type dCDK8 tagged with EGFP at the C-terminus of dCDK8 can rescue the *cdk8* null (*cdk8^K185^*) mutant in the larval-pupal transition [21], we used the same approach and generated transgenic flies carrying a mutated version of the LXXLL motif, from LQKLL to AQKAA of dCDK8, as validated by sequencing (Figure 1a). We then generated transgenic flies with the genotype “*CDK8^AQKAA^-EGFP; cdk8^K185^*” as the dCDK8 Leu-rich motif mutants (the genomic locus of *cdk8* is shown in Figure 1b, see Materials and Methods for details), and the genotype “*CDK8^LQKLL^-EGFP; cdk8^K185^*” as the wild-type dCDK8 rescue line. The genotypes of both of the rescued lines were validated by PCR using the genomic DNA of these final genotypes (Figure 1c). The wild-type dCDK8 (*CDK8^LQKLL^-EGFP*) can rescue the delayed timing for pupariation of the *cdk8^K185^* mutants (Figure 1d; [21]) and the rescued animals form normal pupae at a time similar to the control (Figure 1d,e). However, the LXXLL motif mutants (*CDK8^AQKAA^-EGFP; cdk8^K185^*) die as the third instar larvae and can never pupariate (Figure 1d). This phenotype is more severe than that of the *cdk8^K185^* mutants, which can reach pupal stage with a two-day delay (Figure 1d; [21]). These observations suggest that the LXXLL motif of dCDK8 is essential for its ability to rescue the defects in developmental transition, and dCDK8 with this motif mutated may have dominant-negative or toxic effects in vivo. These data provide further evidence that dCDK8 regulates EcR-dependent gene expression in controlling the larval-pupal transition.

### 2.2. The Kinase Activity of CDK8 is Required for Its Regulation of the Timing of Pupariation

CDK8 is the only subunit of the Mediator complex with enzymatic activities, and CDK8 may modulate transcription through its kinase activity. Alternatively, CDK8 may function together with the other three subunits of the kinase module to block the interactions between gene-specific transcriptional factors and the general transcription apparatus [4,6,20]. Given that both modes of actions of CDK8 are disrupted in *cdk8^K185^* null mutants, it is unclear whether the kinase activity of CDK8 per se is required for regulating the developmental timing.

To test this, we analyzed another mutant allele of *cdk8*: *cdk8^H2480^*. *Cdk8^H2480^* harbors a P154L (CCG to CTG) missense mutation, a highly conserved residue within ATP-binding domains of all eukaryotic CDK8 proteins, and this missense mutation is predicted to disrupt the kinase activity of CDK8 [19,24]. This allele was identified through an EMS screen, and the whole-eye *cdk8^H2480^* mutant clones disrupt visual system wiring without affecting the differentiation of photoreceptor cells [24]. Homozygous mutants of the original *cdk8^H2480^* allele are embryonic lethal [24]. To remove potential second-site mutations of this allele, we outcrossed this allele with the control (*w^1118^*) for several generations and then validated it by sequencing (Figure 1f). After outcrossing, the *cdk8^H2480^* homozygous mutants are pupal lethal rather than embryonic lethal (Figure 1d,e). Importantly, the larval-pupal transition of the *cdk8^H2480^* homozygous mutants is severely delayed compared to the control (Figure 1d), which is similar to the *cdk8* null (*cdk8^K185^*) mutants [21]. We note that the level of CDK8 proteins in *cdk8^H2480^* mutant larvae is similar to the control larvae (Figure 1g). Therefore, these observations suggest that the kinase activity of dCDK8 is required for dCDK8 to regulate the larval-pupal transition.

### 2.3. Generation of Theoretical Structures of Drosophila CDK8 and the Mutations of Human and Drosophila CDK8

Based on the experimental data from *Drosophila*, the point mutation (P154L) in dCDK8 was thought to abolish its kinase activity, while the mutation in the LXXLL motif of dCDK8 was expected to impair the larval-pupal transition for failing to activate the expression of EcR-target genes [21,24]. However, the effect of these mutations on the global and local structures of CDK8 proteins remains unexplored. Thus we sought to further study the dynamics of these theoretical structures of human and *Drosophila* CDK8, as well as the P154L and the LQKLL to AQKAA mutations.

Given that molecular dynamics (MD) simulations have been widely exploited in understanding protein structures [29], to evaluate the structural impacts of these mutations on dCDK8, we took advantage of a computational approach to build a theoretical structure of dCDK8 based on the crystal structure of hCDK8 (PDB ID: 3RGF; [27]). These structures allowed us to build the theoretical structures of hCDK8 and dCDK8 with the mutations either in the region of regulating kinase activity (hCDK8-P154L and dCDK8-P154L), or in the LXXLL motif (hCDK8-AQKAA and dCDK8-AQKAA) where all three leucine residues were mutated to alanine. We have used the identical procedures for the MD simulations of the structures including hCDK8, dCDK8 and theses specific mutations (see below).

### 2.4. Analyses of the Potential Energy, Hydrogen Bond, RMSD and Global Structure of CDK8

MD simulations revealed that the heating phase displayed a gradual increase in temperature with fluctuations, and the temperature at the equilibration phase was approximately 325 K as expected (data not shown). We then calculated the energy profile, including kinetic energy, potential energy, and total energy. The calculations show that dCDK8 has slightly lower potential energy than hCDK8 (*t*-test, *p* < 0.001) (Figure 2a). The P154L mutations of both hCDK8 and dCDK8 have nearly no effect on the potential energy (Figure 2a), while the AQKAA mutations of hCDk8 and dCDK8 slightly decrease the potential energy (*t*-test, *p* < 0.001). We have observed that hCDK8 and the mutations have similar numbers of hydrogen bonds (Figure 2b and Appendix A). *Drosophila* CDK8-P154L has slightly fewer hydrogen bonds than dCDK8 and dCDK8-AQKAA (Figure 2b and Appendix A). We extracted the structures with their respective lowest potential energies, the most stable conformations, and the coordinates are provided in the supplemental files. The backbone root-mean-square deviation (RMSD) plots were generated using the lowest-energy structures as the references. The RMSD plots show conformational changes during the simulations. The data showed a gradual decrease in RMSD for hCDK8 and dCDK8 in the beginning and a gradual increase in the late simulation trajectories (Figure 2c,d). Similar trends were observed for hCDK8-AQKAA and dCDK8-AQKAA (Figure 2c,d). In contrast, hCDK8-P154L and dCDK8-P154L exhibit a continuing decrease in RMSD during nearly entire MD simulations (Figure 2c,d). These results indicate the mutation-induced formational changes.

To understand the effects of the point mutations on the overall structure of CDK8, we have superimposed the structures with the lowest potential energies of either hCDK8 or dCDK8 with their respective mutations. We designate the hCDK8 structure with the lowest potential energy during the entire MD simulations as “hCDK8_LowE”. Similarly, hCDK8-P154L_LowE, hCDK8-AQKAA_LowE, dCDK8_LowE, dCDK8-P154L_LowE and dCDK8-AQKAA_LowE are defined as the structures with the lowest potential energy. We have found that the N-lobe and the C-lobe are well separated in all structures, and the mutations do not dramatically influence the overall structures of N-lobe and C-lobe of hCDK8_LowE (Figure 2e) and dCDK8_LowE (Figure 2f). Several minor changes in these structures (Figure 2e,f) will be further discussed below.

### 2.5. The Analyses of the Hydrogen Bond Occupation between Leu312, Leu315 and Leu316 in the LXXLL Motifs of hCDK8 and dCDK8

As the first step to understanding the differences in local structure between hCDK8 and dCDK8 and potential effects of the mutations on these local structures, we focused on our analyses on the LXXLL motif. There are three hydrogen bonds in the LXXLL motif of the crystal structure of hCDK8 (PDB ID: 3RGF; [27]): one hydrogen bond between L312 and L316 and two hydrogen bonds between L315 and L316 (Figure 3a). During the period of the MD simulations, two additional hydrogen bonds, one between L312 and L315, and another between L312 and L316, were consistently present. The hydrogen bond occupation between L312 and L316 is higher than that between L312 and L315 for all the structures (*t*-test, *p* < 0.05 for each L312-L316 and L312-L315 pair) (Figure 3b). hCDK8 has higher hydrogen bond occupation between L312 and L316 than dCDK8 during the simulations (*t*-test, *p* < 0.01) (Figure 3c). The studies show much higher hydrogen bond occupation between L312 and L315 in hCDK8 during the early steps and has nearly identical occupation of hydrogen bonds during the late steps of the simulations (Appendix A).

The P154L mutation dramatically decreases the hydrogen bond between L312 and L315 in hCDK8 (Figure 3b and Figure 2d), but not in dCDK8 (Figure 3b,e). We did not observe a consistent effect of the P154 mutation on the hydrogen bonds between L312 and L316 for both hCDK8 and dCDK8 (Figure 3b and Appendix A). The two hydrogen bonds between L315 and L316 were broken in the very early steps of the simulation, and they stay unhydrogen-bonded for the majority of the rest trajectories during the entire simulations for hCDK8 and dCDK8 and their mutations.

### 2.6. The Differences in the Pattern of Hydrogen Bonds between Leu312, Leu315 and Leu316 in the LXXLL Motifs of CDK8 Proteins

Interestingly, two hydrogen bonds between L315 and L316 were still observed in hCDK8_LowE (Figure 4a) and dCDK8_LowE (Figure 4b). hCDK8_LowE does not have hydrogen bonds between L312 and L315, and between L312 and L316 (Figure 4a). dCDK8_LowE has a hydrogen bond between L312 and L315, but no hydrogen bond is present between L312 and L316 (Figure 4b). The hCDK8-P154L_LowE (Figure 4c) and hCDK8-AQKAA_LowE (Figure 4d) do not have an obvious effect on the hydrogen bonds in the LXXLL motif. The dCDK8-P154L_LowE switched the hydrogen bond between L312 and L315 to L312 and L316 (Figure 4e), and the hydrogen bond between L312 and L315 was broken in dCDK8-AQKAA_LowE (Figure 4f).

Next, we calculated electrostatic and van der Waals interactions in the LXXLL motif. Our results have revealed that the electrostatic interactions between L315 and L316 contribute to the most stability of the motif (Appendix A), which is consistent with our calculation of the hydrogen bonds. The mutation of Leu to Ala within the leucine-rich motif slightly decreases electrostatic interactions (Appendix A), but dramatically decreases van der Waals interactions (Appendix A) of the motif. There are no differences in the total electrostatic and van der Waals interactions between hCDK8 and dCDK8 (Appendix A). Furthermore, the mutations of P154L and AQKAA of hCDK8 and dCDK8 do not alter dynamics of electrostatic (Appendix A) and van der Waals interactions (Appendix A).

L312, L315 and L316 of the LXXLL motif form a triangle, and the LXXLL has specific binding affinity to nuclear receptors, which regulate a variety of physiological and pathological processes such as carbohydrate and lipid metabolism, as well as inflammation in cell type- and receptor-specific manners [23]. In addition, nuclear receptors also control central pathways impacting a wide range of pathophysiological conditions from cancers to metabolic diseases [30].

### 2.7. The Specific Geometry of the Triangle Formed from C_α_ Atoms of Three Leucine Residues in the LXXLL Motif

Based on the functions of the LXXLL motif, we have hypothesized that the triangle from the C_α_ atoms of three leucine residues in the LXXLL motif has specific backbone geometries compared with the triangles formed from any other three leucine residues in CDK8. The MaxDist and Theta of the LXXLL motifs of the crystal structure and hCDK8_LowE are shown in Figure 5a and Figure 4b. Our study demonstrated that L312, L315 and L316 have shorter MaxDist and larger Theta than the triangles from all other combinations of any three leucine residues for both hCDK8 and dCDK8 (Figure 5c). The mutations of P154L of both hCDK8 and dCDK8 do not alter the geometry of L312, L315 and L316 (data not shown). In addition, the AQKAA mutations of both hCDK8 and dCDK8 do not change MaxDist, but slightly increase Theta of L312, L315 and L316 by average (Figure 5c).

Leucine and isoleucine have similar structures and chemical properties. There is no corresponding IXXII motif, and thus it is unknown whether the geometry of a triangle formed from C_α_ atoms of three isoleucine residues is different from that of a triangle formed from C_α_ atoms of three leucine residues. Since we mutated LXXLL to AXXAA, we also calculated MaxDist and Theta of all combinations of any three isoleucine and any three alanine (Figure 5c). The triangles from any three isoleucine have slightly shorter MaxDist and slightly larger Theta than those formed from three leucine (Figure 5c). The triangles from three alanine have similar MaxDist and larger Theta (Figure 5c). Collectively, these results show that L312, L315 and L316 in the LXXLL motif of CDK8 have the specific pattern of hydrogen bonds and geometries.

### 2.8. The Differences in Hydrogen Bond Occupation and the Effect of CDK8 Mutations on the Hydrogen Bonds between His149 and Asp151, and Between Asp151 and Asn156

There are high hydrogen bond occupations of Asp151 with His149 and with Asn156. P154 is close to P151 in the amino acid sequence. To further understand the difference in local structures between hCDK8 and dCDK8, and effect of the P154L mutation on local structures, we will focus the discussion of hydrogen bond interactions between Asp151 and His149 first, and then the interactions between Asp151 and Asn156.

The analysis of hydrogen bond dynamics shows hCDK8 has higher hydrogen bond occupations between H149 and D151, and between D151 and N156 than dCDK8 (*t*-test, *p* < 0.01 for both H149-D151 and D151-N156 pairs) (Figure 6a–c). The crystal structure of CDK8 has a hydrogen bond between D151 and N156 and no hydrogen bond between H149 and D151 (Figure 6d). The similar pattern of hydrogen bond was observed in dCDK8_LowE (Figure 6e). For hCDK8_LowE, an additional hydrogen bond was created for D151 with H149 (Figure 6f). The hCDK8-P154L mutation slightly increases hydrogen bond occupation between H149 and D151 compared with hCDK8 (Figure 6a and Figure 7a). Moreover, hCDK8 has lower H149-D151 hydrogen bond occupation in the early steps than the late steps of the simulation (Figure 7a).

For dCDK8-P154L, the H149-D151 hydrogen bond occupation stays high in the first two 5-ns simulation (from 0 to 10 ns), but dramatically decreases to nearly zero from the fifth to the tenth 5-ns (from 21 ns to 50 ns) simulations (Appendix A). hCDK8-AQKAA has higher H149-D151 hydrogen bond occupation in the early steps and hCDK8 and hCDK8-AQKAA have similar hydrogen bond occupation in the late step of the simulations (Figure 7a). dCDK8-AQKAA has higher H149-D151 hydrogen bond occupation than dCDK8 (Appendix A). Regarding the hydrogen bond between D151 and N156, hCDK8 has higher hydrogen bond occupations than dCDK8 (Figure 6a). hCDK8-P154L has lower D1451-N156 hydrogen bond occupation than hCDK8 during the early steps, and no difference is observed in the late steps of the simulations (Figure 6a and Figure 7b). hCDK8-AQKAA has lower hydrogen bond occupation between D151 and N156 than hCDK8 (Figure 6a and Figure 7b). We observed no obvious effect in the mutations of dCDK8-P154L and dCDK8-AQKAA on hydrogen bond between D151 and N156 (Appendix A). Furthermore, hCDK8-P154L_LowE has one hydrogen bond between H149 and D151, and one between D151 and N156 (Figure 7c), the same as hCDK8_LowE (Figure 6f). The hydrogen bond between D151 and N156 was broken in hCDK8-AQKAA_LowE (Figure 7d). dCDK8-P154L_LowE (Appendix A) and dCDK8-AQKAA_LowE (Appendix A) created an additional hydrogen bond between D151 and N156 compared with dCDK8_LowE. In summary, the hydrogen bond pattern of Asp151 is different between hCDK8 and dCDK8, and the mutations of P154L or AQKAA alter Asp151-participated hydrogen bonds.

### 2.9. The Effect of hCDK8/dCDK8 Mutations on Hydrogen Bonds between P154 and I157 and Geometry of Triangles Formed from C_α_ Atoms of P(L)154, I157 and D173

Pro154 of hCDK8 crystal structure is hydrogen-bonded to Ile157 (Figure 8a). A highly conserved Asp173 of hCDK8 is predicted to bind phosphate group of ATP, and the D173A mutation inactivates the kinase activity of CDK8 [31,32]. Our previous MD simulations show that D173A has high average potential energy, suggesting unstable of the structure with the D173A mutation [26]. The C_α_ atoms of Pro154, Ile157 and Asp173 form a triangle.

To further analyze the effect of these mutations on the local structures of CDK8, we investigated the geometry of the triangle of P154-I157-D173. As predicted, the triangle of P154-I157-D173 of hCDK8 and dCDK8 has smaller MaxDist and slightly larger Theta than those formed from any combinations of Pro, Ile and Asp (Figure 8b). hCDK8-P154L and dCDK8-P154L have slightly larger MaxDist and smaller Theta than hCDK8 and dCDK8 (*t*-test, *p* < 0.01 for both hCDK8-P154L and dCDK8-P154L) (Figure 8b). Interestingly, the mutations of hCDK8-AQKAA and dCDK8-AQKAA, which are far from P154 in the amino acid sequence, also have slightly larger MaxDist but smaller Theta compared to those of hCDK8 and dCDK8 (*t*-test, *p* < 0.01 for both hCDK8-P154L and dCDK8-P154L) (Figure 8b). The hydrogen bond between P154 and I157 is unstable, and was broken in hCDK8_LowE (Figure 8c) and dCDK8_LowE (Figure 8d) during the simulations. P154 or L154 was found to hydrogen-bond to K153 in hCDK8_LowE (Appendix A), hCDK8-P154L_LowE (Appendix A), hCDK8-AQKAA_LowE (Appendix A), dCDK8_LowE (Appendix A), dCDK8-P154L_LowE (Appendix A), and dCDK8-AQKAA_LowE (Appendix A). Interestingly, P154 or L154 has an additional hydrogen bond: L154 and N181 in dCDK8-P154L_LowE (Appendix A), and W105 and P154 in dCDK8-AQKAA (Appendix A). Taken together, these results suggest that the mutations of P154L and AQKAA modestly alter the local structures around residues 154.

### 2.10. The Clustering Result Shows the Difference between hCDK8 and dCDK8 Structures

Protein structural comparison, unlike the counterpart for protein sequence comparison, is still not a widely used method due to limitations of comparison methods available. We have developed a new structure-based method for comparing protein structures. We have selected ten structures, which are uniformly distributed over all frames during the simulation period, from trajectories of hCDK8, hCDK8-P154L, hCDK8-AQKAA, dCDK8, dCDK8-P154L, dCDK8-AQKAA and then performed the structure comparison.

The clustering result shows that nearly all hCDK8 structures are grouped together (Figure 9a). The same situation is observed for dCDK8 and the mutations of hCDK8 and dCDK8 (Figure 9a). It reveals subtle structure differences between hCDK8 and dCDK8, and between hCDK8, dCDK8 and their mutations (Figure 9a). These structure differences were also observed in Figure 2c,d. The successful protein structure clustering prompted us to identify unique substructures for distinguishing between hCDK8 and dCDK8. Such substructure was found. It is formed by five amino acids (Asp282, Phe283, Arg285, Thr287 and Cys291) (Figure 9b).

Moreover, the substructure constituted from two triangles, Phe283-Arg285-Thr287 (Figure 10a) and Asp282-Arg285-Cys291 (Figure 10b) of hCDK8 have shorter MaxDist (*t*-test, *p* < 0.001 both triangles) and larger Theta (*t*-test, *p* < 0.001 both triangles) than those of dCDK8, suggesting that unique geometries of the substructure can be used to distinguish hCDK8 from dCDK8. The geometries of the substructure of hCDK8 and dCDK8 are presented in Figure 10c,d, respectively.

### 2.11. The Substructure Specifically Belonging to hCDK8 with BAX Was Identified

As a subunit of the Mediator complex, CDK8 can either positively or negatively regulate gene transcription [15,33]. The oncogenic effects caused by gained CDK8 in melanoma and colorectal cancers have spurred an interest to develop CDK8-specific inhibitors for cancer treatment [7,32,34,35,36]. Thus we analyzed the specific interactions between hCDK8 and BAX, one of the CDK8 inhibitors [27].

To study interactions between CDK8 and BAX, we need to optimize the geometry of BAX and calculate partial charges for each atom of BAX. The initial coordinates of BAX were used from the crystal structure (PDB ID: 3RGF), and the partial charges for each atom of BAX calculated using Gaussian 09 are shown in Appendix A. Our clustering results show that hCDK8 has three clusters: hCDK8 without BAX and CycC, hCDK8 with BAX, and hCDK8 with CycC (Figure 11a), indicating subtle conformational changes of hCDK8 due to binding of BAX or CycC. Next, we asked whether there are any unique local structures that are exclusively belonging to hCDK8 with BAX. One substructure, constituted from the C_α_ atoms of three residues: Tyr32, Ala63 and Cyc64 of hCDK8 with BAX, has different geometry from those of hCDK8 and hCDK8-CycC (Figure 11b). Tyr32, Ala63 and CycC64 are close to BAX, although they do not have strong direct contacts with BAX (Figure 11c). To monitor dynamics of conformational changes, we calculated van der Waals and electrostatic interactions between hCDK8 and BAX. Three residues: Lys52, Glu66 and Asp173 have strong interactions with BAX. Among these three residues, Glu66 has the strongest electrostatic and total (sum of van der Waals and electrostatic) interactions with BAX (Figure 12a). Electrostatic interactions of each of the three residues with BAX is stronger than van der Waals interactions of its corresponding residue (Figure 12a). D173 has the strongest van der Waals interaction with BAX among these three residues (Figure 12a). D173 is highly conserved from yeast to man [19], and the D173A mutation is the least stable conformation using the MD simulations [26]. The experimental data showed that the D173A mutation of CDK8 completely inactivates its kinase activity [31,32] while this mutation does not affect the transcription repression function of CDK8 [37]. The triangle constituted from the C_α_ atoms of Lys52, Glu66 and Asp173 of hCDK8 with BAX has slightly shorter MaxDist whereas the smallest Theta of the triangle was found in hCDK8 with CycC (Figure 12b). The representative structure of the triangle, Lys52-Glu66-Asp173 is shown in Figure 12c. These three residues are close to BAX (Figure 12c). These results suggest that conformational changes induced upon binding of BAX may alter kinase activity of hCDK8. Lys52, Ala100 and Asp173 have high percentage of hydrogen bonding with BAX (Figure 12d). In contrast, Glu66 does not participate in hydrogen bonding with BAX.

dCDK8 and hCDK8 are highly conserved. It is not surprising that the residues, Lys52, Glu66, Ala100 and Asp173 of hCDK8 predicted to have strong interactions with BAX, were also found in dCDK8 (Figure 13a). To predict potential binding of BAX to dCDK8, we compared the structures between dCDK8 and hCDK8. Both structures do not have BAX. We found that the triangle Lys52-Glu66-Ala100 and Glu66-Ala100-Asp173 of dCDK8 have larger MaxDist than the corresponding triangles in hCDK8 (Figure 13b). Those triangles have more or less differences in Theta between hCDK8 and dCDK8 (Figure 13b,c), suggesting that BAX will bind hCDK8 and dCDK8 differently. In summary, we identified BAX-induced subtle conformational changes in the domain of hCDK8 that is essential for kinase activity.

## 3. Discussion

The structure of dCDK8 has not been determined by experimentation, making it challenging to understand the structural impacts of different mutations on the overall structure of dCDK8. To address this problem, we have used the MD simulation approach and generated the structure of dCDK8 protein based on the hCDK8 structure, which was determined by crystallography [27]. The simulated dCDK8 structure allowed us to analyze two mutant forms of dCDK8 proteins, P154L and AQKAA, to gain the insights into the potential structural consequences of these mutations. Our MD simulation analyses shed light on structural dynamics of both hCDK8 and dCDK8, as well as the effects of mutations on their local structures.

The three leucine residues in the LXXLL motif, which is often found in cofactors for nucleus receptors, usually posits in a triangle configuration and inserts into a hydrophobic pocket of the interacting nucleus receptors [38,39]. Our simulations of the mutant form of the Leu-rich motif (AXXAA) in CDK8 protein suggest that the overall structure of the CDK8 protein is not significantly altered by this mutation. However, the AXXAA mutation weakly decreases the pairwise electrostatic interactions (Appendix A), but dramatically decreases the van der Waals pairwise interactions between residues 312, 315 and 316 (Appendix A). These changes indicate that the AXXAA mutation may weakly affect the stability of the motif, but may have a strong impact on the interactions between CDK8 and other proteins. Accordingly, these results predict that the energetical changes of the AXXAA motif may disrupt the interactions between dCDK8 and EcR, thereby decreasing the transcriptional activity of EcR target genes. Downregulated expression of EcR target genes may delay the larvae to pupae transition, as we have observed in *cdk8* and *cycC* mutants [21,22]. Our observation that wild-type dCDK8, but not *dCDK8-AXXAA* mutant allele, can rescue the *cdk8* null mutants, provide a key evidence to validate the model that the LXXLL motif is critical for CDK8 to function as a cofactor of EcR in regulating the larval-pupal transition, as we proposed previously [21].

Woven into the complexity of the afore-discussed aspects is the kinase activity of CDK8. The level of CDK8 protein in the CDK8 mutants (*cdk8^H2480^*) is similar to that of the control. The highly conserved Pro154 within the ATP binding domain of CDK8 is expected to be critical for CDK8 activity. It is unlikely that this point mutation (CDK8-P154L) may affect its interaction with CycC, thus the function of the CDK8 module as a physical block between the small Mediator complex and the general transcription apparatus may not be affected in the kinase-dead CDK8 mutants. Yet the kinase-dead CDK8 (*cdk8^H2480^*, CDK8-P154L) mutant larvae display similar effects to the *cdk8* null (*cdk8^K185^*) and *cycC* null (*cycC^Y5^*) mutants, suggesting that the kinase activity of CDK8, instead of merely serving as a physical block, is essential for its function in vivo.

Our analyses show that the P154L mutation does not significantly affect the overall structures of hCDK8 and dCDK8, and that the P154L mutation has little effect on the Leu-rich motif, suggesting that P154L mutation may not affect the interactions between CDK8 and its substrates. However, the P154L mutation alters the hydrogen bond with the surrounding residues. In addition, hCDK8-P154L and dCDK8-P154L have slightly larger MaxDist and smaller Theta than hCDK8 and dCDK8 in the triangle of P154-I157-D173 (Figure 8b), suggesting the change of the environment of D173. Given that D173 is a critical residue for the kinase activity of CDK8 [26], these changes caused by the P154L mutation may also abolish the kinase activity of CDK8. The notion that P154L mutation results in the loss of CDK8 kinase activity is consistent to the severe developmental defects observed in the *dCDK8-P154L* mutants in *Drosophila* ([24] and this work). Thus, our analyses provide a structural explanation to understand how dCDK8-P154L mutation abolishes the kinase activity of dCDK8. Our developmental genetic analyses of the dCDK8-P154L (*cdk8^H2480^*) mutants also show that the kinase activity of CDK8, but not the physical presence of CDK8 per se, is critical for its role in regulating the larval-pupal transition.

These results may have important implications to our understanding of human diseases, as different point mutations of CDK8 have been identified in samples from lung and colorectal cancer patients [18]. To understand the functions of CDK8 and the Mediator complexes, it is essential to identify the direct substrates of CDK8 in vivo, a key challenge for the future.

Taken together, our analyses of the P154L and AXXAA mutants of hCDK8 and dCDK8 proteins provide structural insights into the functional consequences of these mutant forms of CDK8 proteins. These results will help to advance our understanding of the exact biochemical and physiological functions and regulation of CDK8, thereby guiding the future experimentation. CDK8 has a paralog protein, CDK19 (also known as CDK8L), in vertebrates [40]; and CDK8 and CDK19 may have partially redundant functions [15]. Given its relative simplicity and other unique advantages for experimentation, we expect that *Drosophila* will continue providing important insights into the function and regulation of the CKM and the Mediator complex in different developmental and physiological contexts in the future.

## 4. Materials and Methods

### 4.1. Drosophila Stocks and Reagents

All flies were maintained on the standard cornmeal-molasses-yeast (CMY) medium at 25 °C. The *cdk8^H2480^/TM3 Sb Ser* stock was a generous gift from Henri-Marc Bourbon [24]. After four generations of outcrossing, we then replaced the balancer chromosome to *TM6B Tb* (*cdk8^H2480^/TM6B Tb*) to facilitate the analyses during larval and pupal stages.

### 4.2. Validation of cdk8 Mutant and Rescued Line

Purification of genomic DNA and the subsequent PCR and sequencing validation of genotypes were performed by the same methods described previously [21]. For validation of the deletion in *cdk8^K185^* mutants, the following primers were used: cdk8-K185F: 5′-TGTGGGCTGGGATTGTTCTGC, and cdk8-K185R: 5′-ACATCTGGGCTATTGGCTGTATTTTCG. The expected product sizes are 1792 bp for control, 910 bp for *cdk8^K185^* deletion, and 2500 bp for *cdk8^+^-EGFP* insertion. For the verification of the insertion of *cdk8*-tagged with EGFP in the rescued *cdk8* line, the following primers were used: cdk8 5.11: 5′-GCAGCAAATGAACGCTGAG and cdk8 IN-4(EGFP): 5′-TGTATCAGTCTCTCACTTGTACAGCTCGTCCATGCCG (the underlined sequence is the 15 bp overlapping sequence used in the In-Fusion HD Cloning Plus (Takara Bio USA #638910; Mountain View, CA, USA), and the expected product size is 931 bp. In addition, these two primers were used to validate the *cdk8^K185^* deletion: CDK8-K185F: 5′-TGTGGGCTGGGATTGTTCTGC; and CDK8-K185R: 5′-ACATCTGGGCTATTGGCTGTATTTTCG.

### 4.3. Generation of dCDK8 Leucine-Rich Motif Mutant Allele cdk8^AQKAA-EGFP^

Point mutations were introduced into the leucine-rich motif in the dCDK8-EGFP rescue construct (pVALIUM20-dCDK8-EGFP) using the In-Fusion HD system, as described in detail previously [21]. The primers used to generate point mutations are as listed below: CDK8-AQKAA-F: 5′-GCGCAGAAGGCGGCGCTAATGGATCCCAACAAGCGC, and CDK8-AQKAA-R: 5′-CGCCGCCTTCTGCGCCAGGTGAAAGGCCTTGCTGTCTG. The final construct was then inserted into the fly genome on the second chromosome (attP40 site at 25C6) using the PhiC31integrase system using the service provided by Rainbow Genetic Flies (Camarillo, CA, USA).

### 4.4. Analyses of the Timing for the Larval-Pupal Transition and Western Blots

These analyses were performed using the same methods as described in detail previously [21,22]. Briefly, a two-hour collection of embryos was made in bottles with fly food after a couple of pre-collections to purge the females of over-aged embryos. The mid-point was considered as the start point of egg laying. The number of pupariated animals were inspected and counted once every 24 h after egg laying (AEL), and Microsoft Excel was used to plot the pupariation curves.

### 4.5. Molecular Modeling of Human and Fly CDK8

The molecular modeling procedure was based on the methods described previously [26,41]. The sequences were aligned using the ClustalW pairwise alignment algorithm of the Vector NTI software (Life Technologies Corporation, Carlsbad, CA, USA) [42]. The alignment data were imported into MODELLER version 9.16 (San Francisco, CA, USA) [43] to generate the structures of *Drosophila* CDK8 and structures of two point mutations: P154L, L312A/L315A/L316A in the LXXLL motif using the experimentally solved human CDK8 structure (PDB ID: 3RGF, Chain A; [27]) as the template.

### 4.6. Molecular Dynamics (MD) Simulations of Human and Fly CDK8 with Their Point Mutations Using the Amber Package

MD simulations were based on the procedure described by Simmerling et al. [29]. All computations for human and *Drosophila* CDK8, and the mutations including the initial energy minimization, heating, and MD equilibration runs, were performed under fully unrestrained conditions. All calculations used Amber’s all-atom force field (ff14SB) as implemented in Amber 16 software [44]. The SANDER and PTRAJ/CPPTRAJ modules [45] of Amber, respectively, were used for computation and analysis. A total of 1000 steps of initial energy minimization, including 500 steepest descent steps (ncyc = 500) followed by 500 conjugate gradient steps (maxcyc-ncyc) using a large cutoff (cut = 999 angstroms) and non-periodic simulation (ntb = 0), were performed to adjust the structures. To give the system time to adjust as temperature was raised to the production temperature, the minimized system was slowly heated from 0 to 325 Kelvin (K) in seven increments of 50 K over 50 ps (5 ps for the first six steps and 20 ps for the seventh step). The equilibration MD simulations were conducted for a total of 50 nano seconds (ns) at a constant 325 K. The 50-ns simulations were carried out by ten continuing equilibrium runs. The restart file of each run was generated, and it was used as the input for the next equilibrium run. The time for each of the equilibrium runs was 5 ns. The generalized Born implicit solvent model was used in the heating and equilibration phases [46,47]. Structure images were prepared using the Visual Molecular Dynamics (VMD) package (Urbana, IL, USA) [48]. For MM-GBSA, Onufriev’s GB model was selected (igb = 2) as well as appropriate values for SURFTEN and SURFOFF, 0.0072 kcal mol^−1^ Å^−2^ and 0.0 kcal mol^−1^, respectively [49].

### 4.7. Structure Comparison of Human and Drosophila CDK8 and Their Mutations

We have developed a new method for quantifying protein structure similarity. The locations of amino acids in the 3-D structure of a protein are assumed to be represented by the (x, y, z) coordinates associated with C_α_ atoms of amino acids. In the structure-based representation of a protein, each combination of three non-collinear amino acids is used to generate a triangle. All possible triangles, which correspond to $C_{n}^{3}$ combinations for a protein with *n* residues, are represented by vertex labels (amino acids), edge lengths and angles and they collectively provide a structural description of the protein. Then, the vertex labels, edge lengths and angles of each triangle are combined into an integer value through a set of rule-based formulae. We refer to the integer representing a triangle as a key. As a result, the 3-D structure of a protein is represented by a vector of keys (a manuscript is under review). Visualization of our protein structure clustering is based on Average Linkage Clustering [50]. Structural images were prepared using the VMD package [48].

### 4.8. Build of the BAX Prmtop and Inpcrd Files Using the Gaussian 09 and Amber Software Packages

The initial 3-D structure for BAX (4-(4-((((4-chloro-3-(trifluoromethyl)phenyl)amino)carbonyl)amino)phenoxy)-N-methylpyridine-2-carboxamide; PDB ID: 3RGF) [27] was used for Gaussian calculations and MD simulations. To perform MD simulations, we need to build the prmtop and inpcrd files of BAX. It involved several steps: geometry and partial charge calculation, and creation of the BAX prepin, frcmod, lib, prmtop and inpcrd files. The crystal structure of BAX does not have hydrogens. So we added hydrogens to the BAX structure (PDB ID: 3RGF) [27] using GaussView 5 (Semichem Inc., Shawnee Mission, KS, USA). The geometry of BAX was optimized using hf/6-31g(d) [51,52] of Gaussian 09 (Gaussian, Inc., Wallingford, CT, USA) and the Merz-Kollman (MK) [53,54] atomic charges of BAX were calculated using Gaussian 09. The BAX.prepin file was generated by Amber Antechamber module [55] using the optimized geometry and partial charges of BAX calculated from Gaussian 09. The BAX.frcmod file was created from BAX.prepin using the Amber Parmchk module. The xLeap module of Amber 12 [44] was used to build the BAX.lib file. Both the BAX.lib and BAX.frcmod were used to build the BAX.prmtop and BAX.inpcrd files using xLeap. All the BAX.lib, BAX.prepin, BAX.frcmod, BAX.prmtop and BAX.inpcrd files will be available upon request. The solvated hCDK8-BAX and hCDK8-CycC complexes were created in TIP3P water box [56]. Ions were added to neutralize the protein complex using the model of frcmod.ionsjc_tip3p [57].

## Figures and Tables

**Figure 1 ijms-21-07511-f001:**
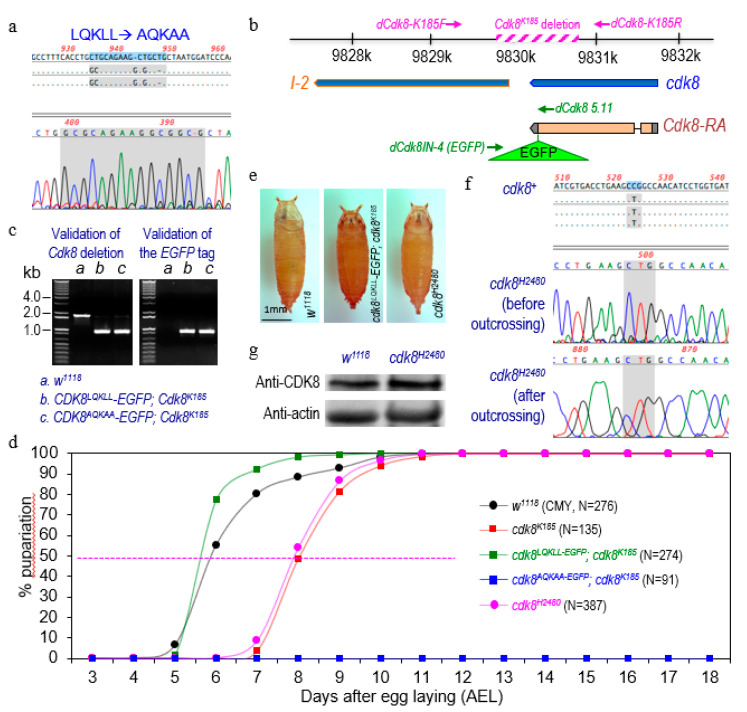
Both the leucine-rich motif and the kinase activity of CDK8 are required for CDK8 to regulate the timing for pupariation. (**a**), Validation of the leucine-rich motif (LQKLL) mutation (AQKAA) by sequencing. (**b**), Wild-type CDK8 (*cdk8^LQKLL^-EGFP*) can rescue the *cdk8* null mutants (*cdk8^K185^*) to adult stage (a rescue pupa is shown), and the kinase-dead CDK8 (*cdk8^H2480^* homozygous) mutants are lethal at the pupal stage. (**c**), Validation of the genotypes of the rescued animals using genomic DNA and PCR. The genotypes: a, *w^1118^*, b, *cdk8^LQKLL^-EGFP*; *cdk8^K185^*, and c, *cdk8^AQKAA^-EGFP; cdk8^K185^*. (**d**), Percentage of pupariated animals with the indicated genotypes raised on the CMY (cornmeal-molasses-yeast) diet. (**e**), Pupal morphology of the indicated phenotypes. (**f**), Validation of the kinase-dead CDK8 (*cdk8^H2480^*) before and after outcrossing by sequencing. (**g**), The levels of dCDK8 proteins in the third instar wandering larvae of the control (*w^1118^*) and *cdk8^H2480^* homozygous mutants; the levels of actin serve as the loading control.

**Figure 2 ijms-21-07511-f002:**
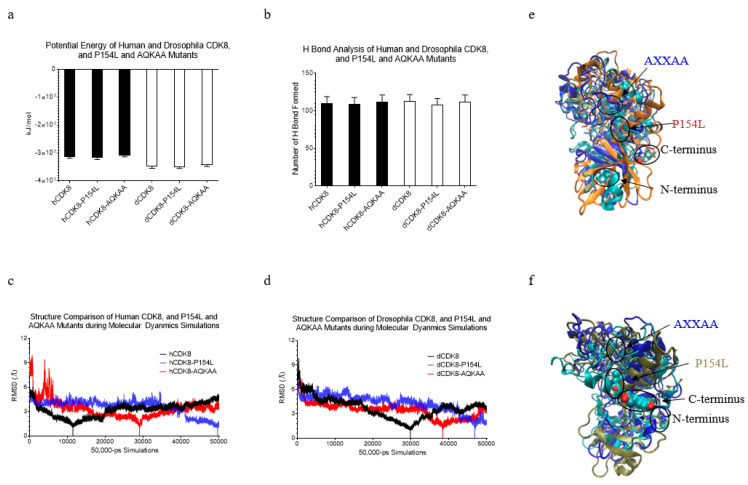
Analyses of the potential energy, hydrogen bond, RMSD and global structure of CDK8 proteins. (**a**), Potential energy profile; (**b**), Numbers of a total hydrogen bond; (**c**), RMSD plot of hCDK8, hCDK8-P154L and hCDK8-AQKAA; (**d**), RMSD plot of dCDK8, dCDK8-P154L and dCDK8-AQKAA; (**e**), Superimposition of three structures using stamp structural alignment function of VMD. Cyan: hCDK8 with the lowest potential (hCDK8_LowE). Orange: hCDK8-P154L with the lowest potential (hCDK8-P154L_LowE). Blue: hCDK8-AQKAA with the lowest potential (hCDK8-AQKAA_LowE); (**f**), Superimposition of three structures used stamp structural alignment function of VMD. Cyan: dCDK8 with the lowest potential (dCDK8_LowE). Tan: dCDK8-P154L with the lowest potential (dCDK8-P154L_LowE). Blue: dCDK8-AQKAA with the lowest potential (dCDK8-AQKAA_LowE); (**a**,**b**), The average and SD are shown in the graph; (**e**,**f**), The N- and C-termini and the mutation sites are labeled.

**Figure 3 ijms-21-07511-f003:**
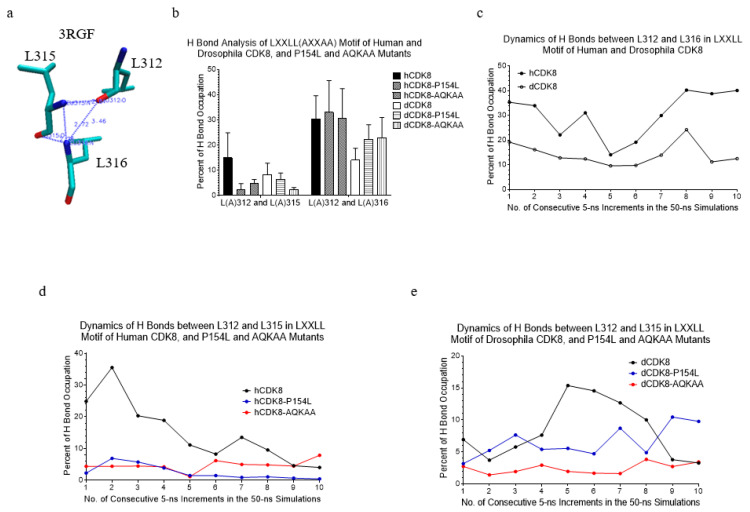
The analyses of the hydrogen bond occupation between Leu312, Leu315 and Leu316 in the LXXLL motifs of hCDK8 and dCDK8. (**a**), Hydrogen bonds of L312, L315 and L316 are shown (PDB ID: 3RGF). The distances of Leu312:O and Leu316:N, Leu315:N and Leu316:N, and Leu315:O and Leu316:N are 2.84 Å, 2.86 Å, and 2.26 Å respectively; (**b**), The hydrogen bond occupations between L(A)312 and L(A)315, and between L(A)312 and L(A)316 of hCDK8, dCDK8 and their mutations were calculated and are present (mean ± SD); (**c**), The dynamics of hydrogen bonds between L312 and L316 in the LXXLL motifs of hCDK8 and dCDK8; (**d**), The dynamics of hydrogen bonds between L312 and L315 in the LXXLL motifs of hCDK8, hCDK8-P154L and hCDK8-AQKAA; (**e**), The dynamics of hydrogen bonds between L312 and L315 in the LXXLL motifs of dCDK8, dCDK8-P154L and dCDK8-AQKAA.

**Figure 4 ijms-21-07511-f004:**
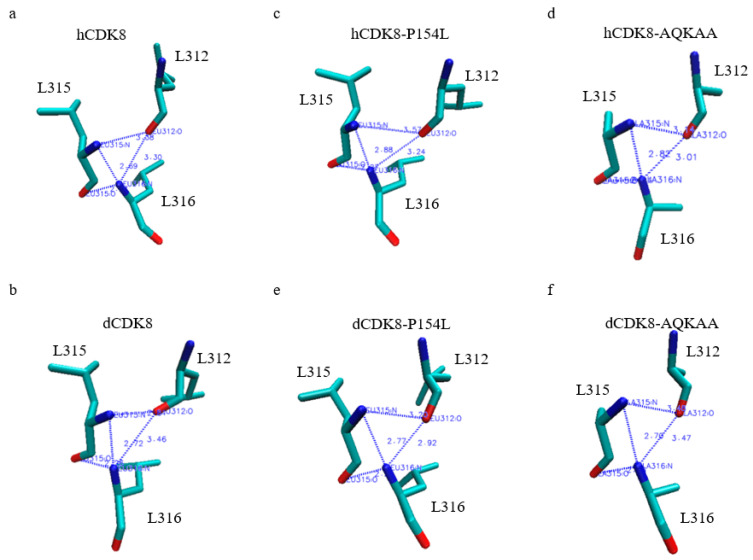
The differences in the pattern of hydrogen bonds between Leu312, Leu315 and Leu316 in the LXXLL motifs of CDK8 proteins. (**a**), hCDK8_LowE: the distances of Leu315:N and Leu316:N, and Leu315:O and Leu316:N are 2.69 Å, and 2.27 Å respectively; (**b**), dCDK8_LowE: The distances of Leu312:O and Leu315:N, Leu315:N and Leu316:N, and Leu315:O and Leu316:N are 2.81 Å, 2.72 Å, and 2.26 Å respectively; (**c**), hCDK8-P154L_LowE: the distances of Leu315:N and Leu316:N, and Leu315:O and Leu316:N are 2.88 Å, and 2.27 Å respectively; (**d**), hCDK8-AQKAA_LowE: the distances of Leu315:N and Leu316:N, and Leu315:O and Leu316:N are 2.82 Å, and 2.24 Å respectively; (**e**), dCDK8-P154L_LowE: The distances of Leu312:O and Leu316:N, Leu315:N and Leu316:N, and Leu315:O and Leu316:N are 2.92 Å, 2.77 Å, and 2.25 Å respectively; (**f**), dCDK8-AQKAA_LowE: the distances of Leu315:N and Leu316:N, and Leu315:O and Leu316:N are 2.70 Å, and 2.26 Å respectively.

**Figure 5 ijms-21-07511-f005:**
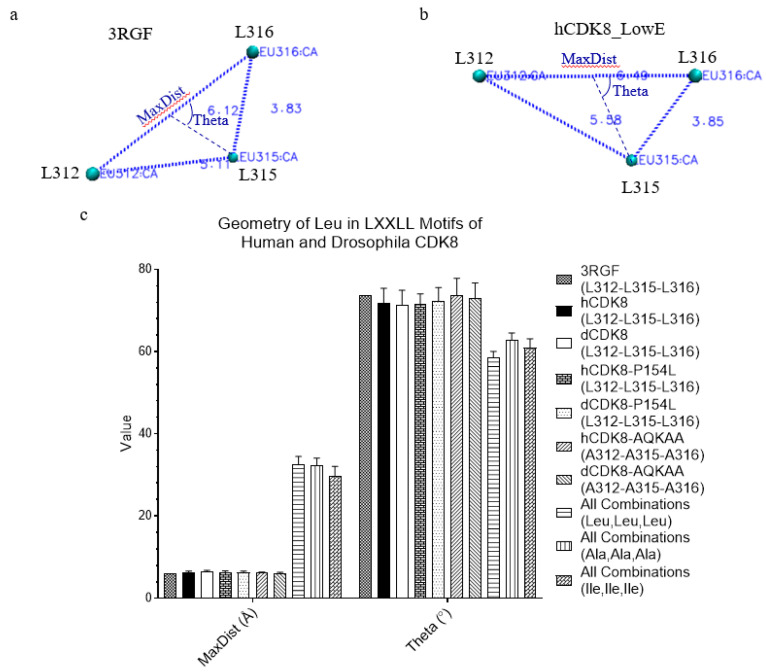
The specific geometry of the triangle formed from C_α_ atoms of three leucine residues in the LXXLL motif. (**a**), The triangle constituted from Leu312, L315 and L316 is present (PDB ID: 3RGF); (**b**), The triangle constituted from Leu312, L315 and L316 of hCDK8_LowE is present; (**c**), The MaxDist and Theta values of the triangles were calculated and are present (mean ± SD). MaxDist is defined as the distance of the longest edge of a triangle. MaxDist is Euclidean distance calculated using the coordinates of two vertices. Theta is the angle that is less than 90° between the line from the midpoint of the longest edge to the vertex which other two edges intersect and half of the longest edge. Theta is calculated using the coordinates of three vertices; a-b, Atom name, distance, MaxDist and Theta are labelled.

**Figure 6 ijms-21-07511-f006:**
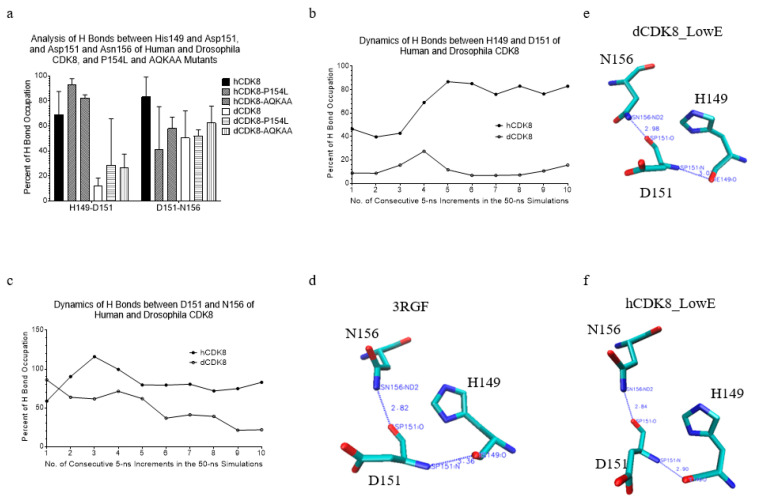
The differences in hydrogen bond occupation between His149 and Asp151, and between Asp151 and Asn156 of hCDK8 and dCDK8. (**a**), The hydrogen bond occupations between H149 and D151, and between D151 and N156 of hCDK8, dCDK8 and their mutations were calculated and are present (mean ± SD); (**b**), The dynamics of hydrogen bonds between H149 and D151 of hCDK8 and dCDK8; (**c**), The dynamics of hydrogen bonds between D151 and N156 of hCDK8 and dCDK8; (**d**), Hydrogen bonds among H149, D151 and N156 are shown (PDB ID: 3RGF). The distance between Asp151:O and Asn156:D2N is 2.82 Å; (**e**), dCDK8_LowE: the distance between Asp151:O and Asn156: ND2 is 2.84 Å; (**f**), hCDK8_LowE: the distances between His149:O and Asp151:N, and between Asp151:O and Asn156: ND2 are 2.90 Å and 2.84 Å respectively.

**Figure 7 ijms-21-07511-f007:**
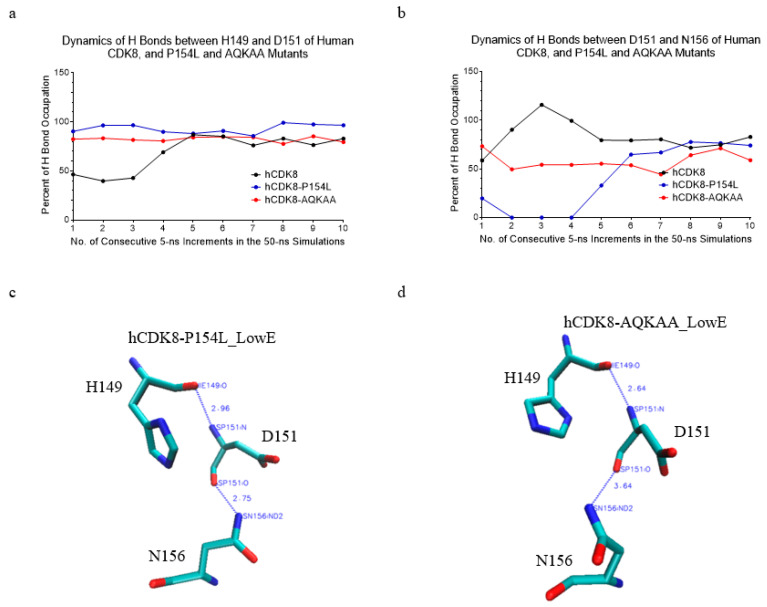
The effect of hCDK8 mutations on hydrogen bonds between His149 and Asp151, and between Asp151 and Asn156. (**a**), The dynamics of hydrogen bonds between H149 and D151 of hCDK8, hCDK8-P154L and hCDK8-AQKAA; (**b**), The dynamics of hydrogen bonds between D151 and N156 of hCDK8, hCDK8-P154L and hCDK8-AQKAA; (**c**), hCDK8-P154L_LowE: the distances between His149:O and Asp151:N, and between Asp151:O and Asn156:ND2 are 2.96 Å and 2.75 Å respectively; (**d**), hCDK8-AQKAA_LowE: the distance between His149:O and Asp151:N is 2.64 Å.

**Figure 8 ijms-21-07511-f008:**
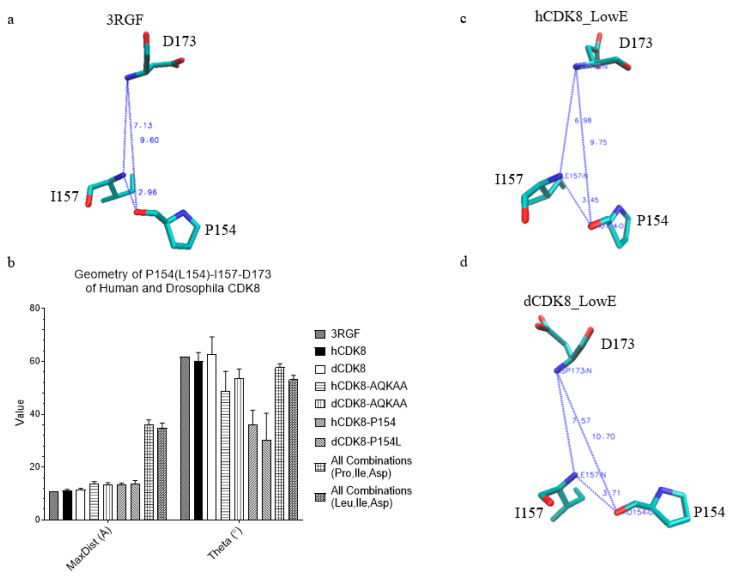
The effect of hCDK8/dCDK8 mutations on hydrogen bonds between P154 and I157 and geometry of triangles formed from C_α_ atoms of P(L)154, I157 and D173. (**a**), PDB ID: 3RGF: the distance between P154:O and I157:N is 2.96 Å; (**b**), The MaxDist and Theta values of the triangles from C_α_ atoms of P(L)154, I157 and D173 were calculated and are present (mean ± SD); (**c**,**d**), The shortest distances between any two amino acids of P154, I157 and D173 of hCDK8_LowE (**c**) and dCDK8_LowE (**d**) are shown.

**Figure 9 ijms-21-07511-f009:**
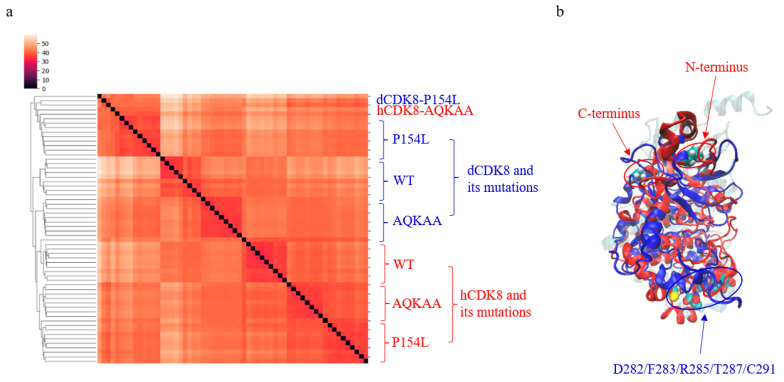
Clustering of hCDK8, dCDK8 and their mutations and the identification of the specific substructures for hCDK8 and dCDK8. (**a**), Structure-based clustering of hCDK8, dCDK8 and their mutations. The percent of structure difference is indicated; (**b**), Superimposition of three structures used stamp structural alignment function of VMD (Cyan and Glass1: CDK8 (PDB ID: 3RGF), Red: hCDK8_LowE, Blue: dCDK8_LowE). The N- and C-termini of hCDK8 and the substructure of dCDK8 are circled and labeled.

**Figure 10 ijms-21-07511-f010:**
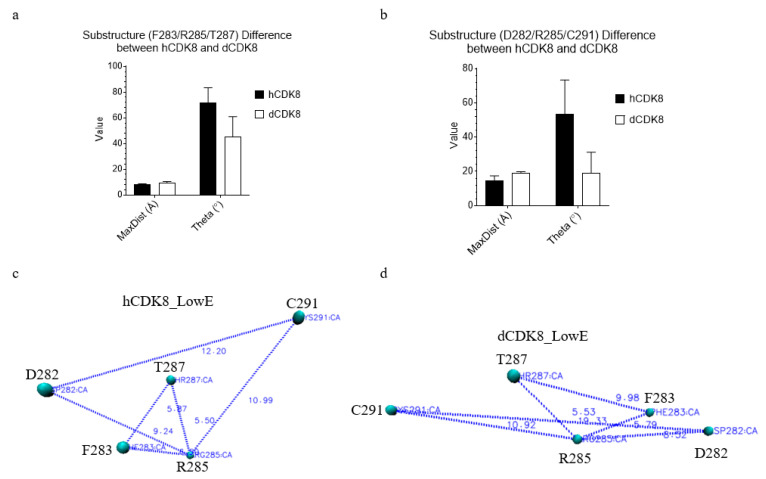
The geometry of the substructure can be used to distinguish hCDK8 from dCDK8. (**a**), The MaxDist and Theta values of the triangle from C_α_ atoms of Phe283, Arg285 and Thr287 of hCDK8 and dCDK8 were calculated and are present (mean ± SD); (**b**), The MaxDist and Theta values of the triangle from C_α_ atoms of Asp282, Arg285 and Cys291 of hCDK8 and dCDK8 were calculated and are present (mean ± SD); (**c**,**d**), The representative substructures are shown for hCDK8 (**c**) and cCDK8 (**d**). The amino acid and their positions are labeled.

**Figure 11 ijms-21-07511-f011:**
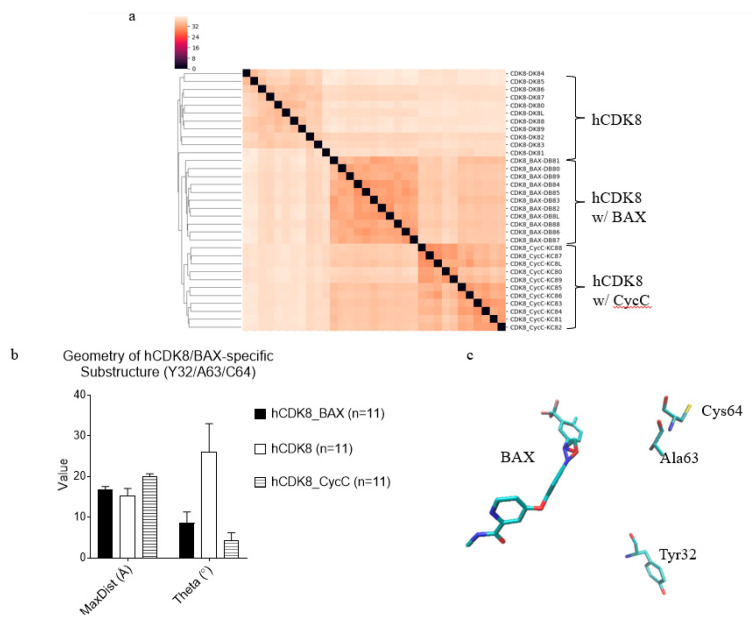
Clustering of hCDK8, hCDK8 with BAX and hCDK8 with CycC, and identification of BAX-induced conformational changes in hCDK8. (**a**), Structure-based clustering of hCDK8 structures. The percent of structure similarity is indicated; (**b**), The MaxDist and Theta values of the triangle from C_α_ atoms of Tyr32, Ala63 and Cyc64 of hCDK8_BAX, hCDK8 and hCDK8_CycC were calculated and are present (mean ± SD); (**c**), The representative substructure, Tyr32-Ala63-Cys64 with BAX of hCDK8 with the lowest potential energy extracted from the trajectories, is shown.

**Figure 12 ijms-21-07511-f012:**
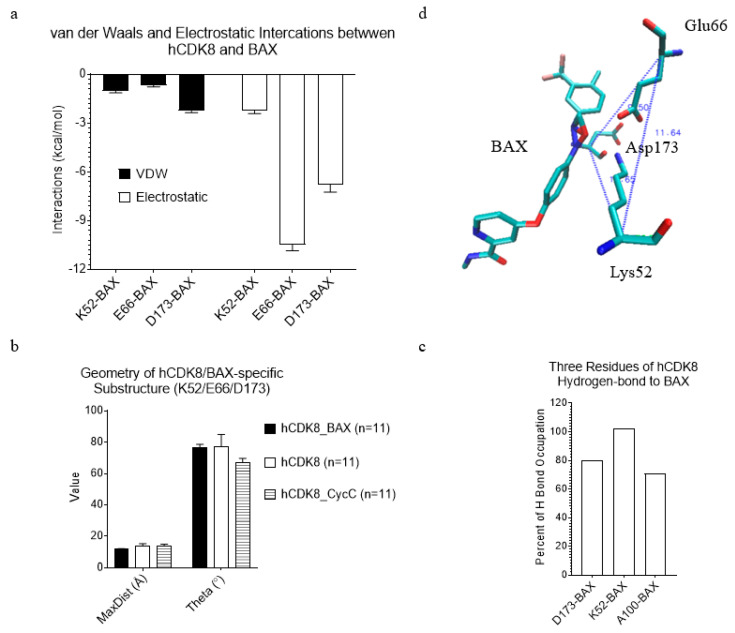
The residues of hCDK8 that have strong interactions with BAX were identified. (**a**), The van der Waals and electrostatic interactions between each of Lys52, Glu66 and Asp173, and were calculated using MM-GBSA method (mean ± SEM); (**b**), The MaxDist and Theta values of the triangles were calculated and are present (mean ± SD); (**c**), The representative substructure, Lys52-Glu66-Asp173 with BAX of hCDK8 with the lowest potential energy extracted from the trajectories, is shown; (**d**), The percent of hydrogen bond occupation was calculated and is present.

**Figure 13 ijms-21-07511-f013:**
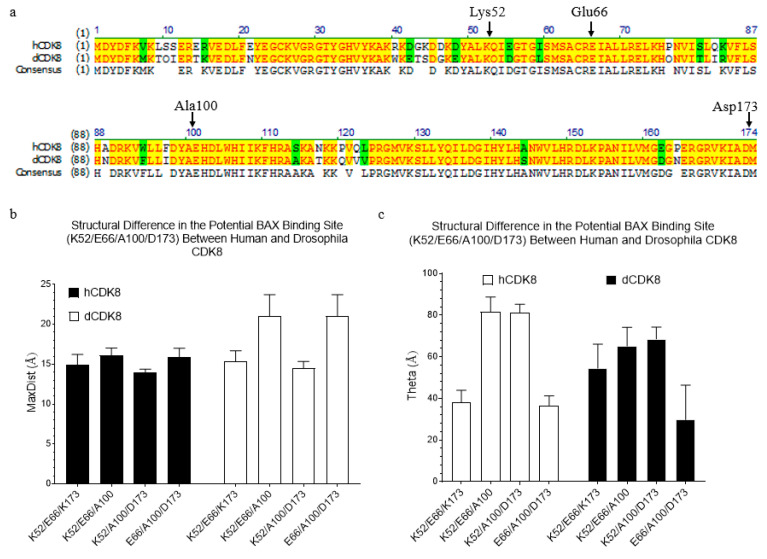
The sequence alignment of hCDK8 and dCDK8, and identification of substructure difference in the potential BAX binding site between hCDK8 and dCDK8. (**a**), The sequence alignment of hCDK8 and dCDK8 was performed using Clustal W algorithm. The residues in the BAX binding site are labeled; (**b**,**c**), The MaxDist (**b**) and Theta (**c**) values of the triangles were calculated and are present (mean ± SD).

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
