# Peer review of "All-Atomic Molecular Dynamic Studies of Human and Drosophila CDK8: Insights into Their Kinase Domains, the LXXLL Motifs, and Drug Binding Site"

_ijms, 2020, doi:10.3390/ijms21207511_

Round 1
Reviewer 1 Report
The Drosophila data added to the revised manuscript provide a convincing proof that dCDK8-AQKAA and dCDK8-P154L mutants are unable to fulfill normal dCDK8 physiological functions. The addition of MD analysis of hCDK8/BAX shows some interesting predictions of target-drug specific interactions. However, the paper requires a number of revisions to deliver its messages more clearly. Below are some questions and suggestions:
- All theoretical structures of hCDK8s and dCDK8s done with MD simulation were based on the crystal structure of hCDK8 (3RGF). There are over 20 hCDK8 crystal structures available in PDB. Do theoretical structures built from different crystal structures differ? If yes, will they reveal similar changes between hCDK8 and dCDK8, or wild-type vs mutant?
- The figure legend of Fig 2 states that the bar graph shows the average and SD - of what? Multiple simulation results? What is n? Should some statistics be used to draw the conclusion? What are the prediction on protein properties based on the MD simulation? For example: dCDK8-AQKAA has less potential energy than dCDK8, does it suggest that dCDK8-AQKAA is less conformationally stable than dCDK8? CDK8-P154L mutant shows different RMSD dynamics from wild-type CDK8 during MD simulations. What does it suggest?
- Section 3. 4 (Fig 3): Does H-bond occupanchy percentage predict the strength of interaction between the two amino acids? What conclusion or prediction can we draw based on the data (such as higher hydrogen bond occupancy between L312 and L315)?
- Fig. 4: Please increase the font size and clarity of the labels of amino acids and H-bond interactions in the figures. It would be better to label substructure plots with their names (hCDK8, dCDK8, hCDK8-AQKAA, etc.) at the top to make the figure more reader-friendly. For Fig. 4f, how can we tell that the hydrogen bond between L312 and L315 was broken? There is a dotted line between them showing interaction.
- Fig.5 and Fig 8: The MaxDist and Theta values of the triangles formed from C-alpha of three amino acids seem to be an interesting predictive parameter for specific substructures. Are the values of “All combinations” calculated from all combinations of any three Leu/Ile/Ala of simulated hCDK8 structure or all three hCDK8 structures (hCDK8 and hCDK8 mutants)? The plot should compare the “all combination” values of each hCDK8 structure (hCDK8, hCDK8-AQKAA and hCDK8-P154L separately). In addition, the MaxDist and Theta values of the crystal structure (3RGF) should be included for comparison.
- Fig 9: What values do the columns of the clustering plot represent? What are the five replicates of each CDK8 structure? The labels are very hard to read. Please increase the resolution.
- The messages of Fig 10a-c are unclear. No difference can be seen between hCDK8 and dCDK8 on these hCDK8- or dCDK8- specific substructures. And in Fig 10g-i, the species-specific substructures should be compared with the ones at the same location in the other species (For example, hCDK8 R285/S286/T287 vs dCDK8 R285/N286/T287).
- It is good to see the authors add simulation analysis of hCDK8 with an inhibitor (BAX) and cyclin partner (CycC). However, we cannot see any difference among these three simulated structures in Fig. 11. Same questions are raised here as in Fig. 9.
- The analysis done in Fig 12 should be applied to dCDK8/BAX to see if BAX has strong interactions with the residues in dCDK8. Combined with results in Fig. 13, authors should make a clear prediction whether BAX strongly interacts with dCDK8 in the same pattern as it interacts with hCDK8.
- The discussion section provide a clear summary about their MD simulation analysis and tell readers what useful information can be acquired from this study. How do the findings help us understand why the AQKAA mutant and P154L mutant do not function like the wild-type? What are the insights from the difference identified between hCDK8 and dCDK8 MD simulation analysis? Can we predict whether an hCDK8 inhibitor will work on dCDK8?
Reviewer 2 Report
The paper has been enriched with experimental study and with several corrections that improved the quality of the manuscript. However, I have some issues:
Section 2.4: a very short description of the methodology is required. If in the text is too repetitive, authors could add it in the SI
Lines 147-148: the continuous equilibrium runs. It should be confusing saying that authors did continuous runs. Are these runs part of a longer run or are they separated runs? If so, how authors coupled the previous run with the folvogin one? Did they used the same velocities and force to the following run?
Lines 181-182: which concentration of ions?
All figures are in poor quality. I suggest to improve the image quality. Moreover, In figures containing the triangle measurements, the amino acids labels and the numbers are difficult to read because they are superimposed to the images (such as the Figures in the SI).
line 296: "if" in "of"
Lines 274-277: regarding the simulations length: looking at the RMSD plots probably 50ns are not enough. However, changes are a not so big, but I suggest in the future to simulate until the RMSD is stable, meaning a complete rearrangement of the protein, in particular if they are mutants
Figure 2 and 9: images of the proteins are not clear. Please, add some labels of peculiar region of the proteins, such as the N- and C- terminal or the muted amino acids.
In general, the triangle method is interesting, but it seems to me that some of the triangles could be refereed to specific secondary structures such as alpha helices. Is this assumption correct? Is there any way to relate the values of the triangles (theta etc.) to the local secondary structure?
I suggest in Figures S6 and S7 to change the y scale of the graph because there are not differences in the data
Author Response
Please see the attachment.

This manuscript is a resubmission of an earlier submission. The following is a list of the peer review reports and author responses from that submission.
Round 1
Reviewer 1 Report
In this paper, authors used classical molecular dynamics to study the main features of human and drosophila CDK8 and some mutants. The goal of this paper is not fully explained and the discussion do not completely address the issues aride in the introduction.
In the methods section authors used 5-ns stages in the production runs. Is it the time at which the measurements are taken? Moreover it seems that authors performed "ten 5-ns stages" of simulations. I suggest to use the simulation time in ns to define the intervals of both production runs and heating.
It should be useful to add the details of the simulation that are missing, such as the solvent and ions treatment. In the paper authors say that they used the generalized Born solvent model in the heating and equilibration phase. Some additional details are required so the study would be easier to be reproduced by other scientists.
In sections 3.2: line 137. abrupt changes of temperature. How big are the changes? Are they only referred to the jumps in the temperature during heating or are they related to something else? Do the systems tolerate well the jumps in the temperature?
line 144: "number if " in "number of"
line 146: in supplementary files I do not see the coordinates. Did authors will load in a different file at the end of the paper evaluation?
lines 149-152: why these continuing decrease/increase in the RMSD? It seems that simulations did not reach a stable configuration. In few cases an elongation of the simulation will be required in order to see if these changes are normal protein fluctuations or not.
Authors should define how the MaxDist and Theta are computed.
line 272: "hydrogen bind" in "hydrogen bond"
line 297-298: the first 5ns and fifth to ten 5ns should be referred to the simulation time, as previously said.
Lines 314 and 336-337: the mutations alter and modestly alter the hydrogen bonds. The extent of the changes is not clear.
Lines 348-350: why random configurations and not the lowest energy configurations?
Line 351: which clustering method did authors used? Results of the clustering procedure will be explained better. It is not clear why to perform this analysis.
Reviewer 2 Report
In this MS, authors used molecular dynamic simulation method to analyze protein structure of wild-type human and drosophila CDK8 proteins as well as their P154L and LXXLL motif mutants. Based on simulation analysis, authors concluded that there are some structural differences between human and drosophila CDK8 and between wild-type and mutant proteins. This study is a potentially useful analysis that could be combined with experimental results testing the predictions of this simulation but no experimental data are included in this paper.
The results of the analysis concur with a priori expectations. The aa sequences of hCDK8 and dCDK8 differ by ~5-10% and it is not surprising that the ortholog-specific substructures were predicted at places where aa sequences differ between hCDK8 and dCDK8. Most of the structural variations mentioned in the paper are focused on the regions of mutations and do not provide much useful information to help us understand functional differences between hCDK8 and dCDK8.
The paper would be more interesting if the authors could apply the same computational analysis based on hCDK8 structures in complex with various selective CDK8/19 inhibitors (PDB files are available) and test whether dCDK8 is able to form similar drug-target interaction with these inhibitors. This would be important for being able to use pharmacological CDK8/19 inhibitors for Drosophila studies.
There is no clear explanation on how to interpret the results presented in this paper. It is impossible for a reader outside of field to link the simulation data to protein structures. What do potential energy profile, total hydrogen bond, RMSD values mean and how do these numbers correspond to protein structures? What is the difference between the results from early steps and late steps of simulation? Authors need to explain these issues in the paper.
